# Comparison of Eight Commercially Available Faecal Point-of-Care Tests for Detection of Canine Parvovirus Antigen

**DOI:** 10.3390/v13102080

**Published:** 2021-10-15

**Authors:** Julia Walter-Weingärtner, Michèle Bergmann, Karin Weber, Uwe Truyen, Cosmin Muresan, Katrin Hartmann

**Affiliations:** 1Clinic of Small Animal Medicine, Centre for Clinical Veterinary Medicine, LMU Munich, Veterinaerstrasse 13, 80539 Munich, Germany; N.Bergmann@medizinische-kleintierklinik.de (M.B.); karin.weber@lmu.de (K.W.); hartmann@lmu.de (K.H.); 2Institute of Animal Hygiene and Veterinary Public Health, University of Leipzig, An den Tierkliniken 1, 04103 Leipzig, Germany; truyen@vetmed.uni-leipzig.de; 3Faculty of Veterinary Medicine, University of Agricultural Sciences and Veterinary Medicine, Str. Calea Manastur 3–5, 400372 Cluj Napoca, Romania; cosmin.muresan@usamvcluj.ro

**Keywords:** CPV, parvovirosis, diagnosis, POCT, in-house test, sensitivity, specificity

## Abstract

A real-time polymerase chain reaction (qPCR) is considered the gold standard for the laboratory diagnosis of canine parvovirus (CPV) infection but can only be performed in specialized laboratories. Several point-of-care tests (POCT), detecting CPV antigens in faeces within minutes, are commercially available. The aim of this study was to evaluate eight POCT in comparison with qPCR. Faecal samples of 150 dogs from three groups (H: 50 client-owned, healthy dogs, not vaccinated within the last four weeks; S: 50 shelter dogs, healthy, not vaccinated within the last four weeks; *p* = 50 dogs with clinical signs of CPV infection) were tested with eight POCT and qPCR. Practicability, sensitivity, specificity, positive (PPV) and negative predictive values (NPV), as well as overall accuracy were determined. To assess the differences between and agreement among POCT, McNemar’s test and Cohen’s Kappa statistic were performed. Specificity and PPV were 100.0% in all POCT. Sensitivity varied from 22.9–34.3% overall and from 32.7–49.0% in group P. VetexpertRapidTestCPVAg^®^ had the highest sensitivity (34.3% overall, 49.0% group P) and differed significantly from the 3 POCT with the lowest sensitivities (Fassisi^®^Parvo (27.7% overall, 36.7% group P), Primagnost^®^ParvoH+K (24.3% overall, 34.7% group P), FASTest^®^PARVOCard (22.9% overall, 32.7% group P)). The agreement among all POCT was at least substantial (kappa >0.80). A positive POCT result confirmed the infection with CPV in unvaccinated dogs, whereas a negative POCT result did not definitely exclude CPV infection due to the low sensitivity of all POCT.

## 1. Introduction

Canine parvovirus (CPV) is a common enteric virus in dogs. It emerged from the feline panleukopenia-like virus (FPV-like), CPV-2, in the 1970s, causing an acute haemorrhagic enteritis [1,2,3]. In the following years, the original type CPV-2 was replaced by two antigenic variants: CPV-2a and CPV-2b [2,4,5]. In 2000, a third antigenic type, CPV-2c, was identified which also spread worldwide in the meantime [6,7]. In this paper, the term CPV is used for all canine field and vaccination parvovirus strains (CPV-2a, CPV-2b and CPV-2c). CPV is highly contagious [8] and causes an often fatal disease, especially in puppies. Clinical signs and laboratory findings include diarrhea and vomiting [9], dehydration, as well as leukopenia [10]. Complications, such as bacterial translocation with consecutive septicemia, systemic inflammatory response syndrome (SIRS), hypercoagulability and multiorgan disfunction, can occur, causing high mortality rates in untreated dogs [10,11].

In order to isolate infected patients and to provide an immediate adequate therapy for dogs with this life-threatening infection, a rapid diagnosis is essential. Thus, reliable diagnostic methods to detect CPV infection are highly important. A polymerase chain reaction (PCR) shows the highest sensitivity compared to traditional methods, such as haemagglutination or virus isolation [12]. A quantitative real-time (q)PCR is considered the gold standard to diagnose CPV infection [12,13,14]. However, qPCR can only be performed in specialized laboratories which delays the diagnosis.

As infected dogs shed high amounts of CPV in their faeces [15], several point-of-care tests (POCT) are commercially available for the detection of the CPV antigen in-house. These POCT are based on an enzyme-linked immunosorbent assay (ELISA) or immunomigration technology and detect CPV antigen in faeces within a few minutes.

For test performance, sensitivity is the most important parameter in this scenario to identify and isolate all the affected dogs before infecting other contact animals. Recent studies showed a high specificity of some of these POCT with over 95.1% [14,16], but sensitivity was low, varying between 15.8% and 80.4% compared to the gold standard qPCR [12,14,16,17]. However, only a few POCT (IDEXX SNAP^®^ Parvo, FASTest^®^ parvo strip, and Witness^®^ parvo card) were evaluated in independent studies so far and studies comparing the tests have yet to be performed.

Therefore, the aim of the present study was to evaluate eight commercially available POCT for the detection of the CPV antigen in the faeces of dogs in comparison with the reference standard qPCR. Practicability, sensitivity, specificity, positive (PPV) and negative predictive values (NPV), as well as overall accuracy, were evaluated.

## 2. Materials and Methods

### 2.1. Faecal Samples

A total of 150 faecal samples from three groups of dogs were included in the study. Ages of the dogs (see Appendix A) were from 4 weeks up to 13 years (median age of 3 years). Group H included 50 healthy, client-owned dogs (age: 1–13 years, median: 5 years). Dogs were only included in this group if there were no abnormalities in physical examination and history and the dogs had not been vaccinated against CPV within the last 4 weeks. Group S consisted of 50 shelter dogs (age: 7 months–12 years, median: 7 years), that were also healthy in their physical examinations and not vaccinated against CPV within the last 4 weeks. These dogs were considered to have a high risk of CPV infection because of high population density and stress. Group P included 50 dogs (age: 4 weeks–2 years, median: 3 months) which were presented to veterinary hospitals with suspicion of CPV infection. The dogs of group P had to meet at least 3 of the following criteria: diarrhea, vomiting, bad general condition, fever, neutropenia and/or incomplete vaccination status according to the WSAVA vaccination guidelines [18]. Vaccination status was known for 28 dogs of group P. Three dogs were vaccinated against CPV within the last 4 weeks, 25 dogs were not vaccinated against CPV within the last 4 weeks. For the remaining 22 dogs, vaccination status was unknown. Vaccination status of the dogs of group P is presented in Appendix A. All faecal samples were stored at −80 °C until further processing. The study protocol for this study was approved by the ethical committee of the Centre for Veterinary Clinical Medicine, LMU Munich, Germany (reference number 53-09-09-2015).

### 2.2. Real-Time PCR, Faecal CPV Load

DNA from faecal samples was extracted using the QIAamp DNA Stool Mini Kit (Qiagen) according to the manufacturers’ recommendations. QPCR for detection and quantification of faecal CPV DNA was performed as described by Streck and colleagues (2013) [19] by a person blinded to the results of the POCT. All samples were run in duplicates. To compare results, the number of DNA copies/template was converted to the number of copies/g faeces, based on the individual sample weight.

### 2.3. Virus Culture

Virus cultures were performed for all qPCR-positive faecal samples. Two hundred milligram of faeces were suspended in 2.0 mL phosphate-buffered saline (pH 7.2), centrifuged at 3000× *g* for five minutes and the supernatant was filtered through a 0.22 µm syringe filter. One hundred microlitres of these filter suspensions were used to inoculate Crandell Rees feline kidney cells maintained in Dulbecco’s medium (Biochrom) supplemented with 5% fetal calf serum (Sigma Aldrich), 1% non-essential amino acids (Biochrom) and 1% penicillin-streptomycin (Biochrom). Cultures were incubated at 37 °C, 5% CO_2_. After seven days of incubation, each culture was subcultured. 

### 2.4. Point-of-Care Tests

All faecal samples were analyzed with the eight POCT detecting CPV antigen (details in Table 1) by the same author (J.W.-W.) who was blinded to the results of the qPCR. Samples were only tested once with each of the POCT.

POCT were performed according to the manufacturers’ instructions. Testing delivered results after 5–10 min.

The IDEXX SNAP^®^ Parvo is an ELISA, in which a conjugate (an enzyme-labeled antibody) forms an immune complex with the antigen in the sample and matrix-bound antibodies. Subsequently, a washing step removes unbound debris and unreacted conjugate. Finally, a substrate-based enzymatic reaction generates a blue point in case of a positive test result [20]. All other POCT are lateral flow immunoassays, using immunomigration technologies. In this technology, the antigen in the sample reacts with mobile, monoclonal anti-CPV antibodies conjugated with gold particles. After migration along a nitrocellulose membrane, the antigen antibody complexes are captured by fixed anti-CPV monoclonal antibodies creating a test line in case of a positive result.

### 2.5. Data Analysis

Test results of the eight POCT were compared with results of the qPCR. Practicability, difficulties in test result interpretation, sensitivity (true positive rate), specificity (true negative rate), negative predictive value (NPV) (proportion of predicted negatives that were true negatives), positive predictive value (PPV) (proportion of predicted positives that were true positives) and overall accuracy (OA) (probability that a dog will be correctly classified by the tests; sum of true positives plus true negatives divided by the total number of dogs tested) were calculated and used for comparison of test performances. Sensitivity was considered as most important parameter. The differences between the POCT performance were tested for statistical significance using Cochran’s Q omnibus test at the group level. In case of significant results, pairwise comparison was performed using McNemar’s test to determine significant differences in sensitivity of the POCT. Cohen’s Kappa statistic was performed to assess agreement of the results among the POCT. Values < 0 indicated poor agreement, 0.00–0.20 slight, 0.21–0.40 fair, 0.41–0.60 moderate, 0.61–0.80 substantial, and 0.81–1.00 almost perfect agreement [21]. Logistic regression was performed to assess the detection probability of each POCT as a function of the virus load. Log transform was applied to the number of virus copies prior to regression analysis. The regression results were presented along with the decision threshold of virus copies/g faeces corresponding to a 50% detection probability. Error bounds with 95% confidence were calculated based on the fit statistics. Virus loads in qPCR-positive samples were statistically compared between loads in the virus culture-positive and loads in virus culture-negative using Mann–Whitney U Test. Results were being considered significant for *p* < 0.05. All statistical analyses were performed with Matlab (R2020b, MathWorks, Natick, MA, USA).

## 3. Results

### 3.1. Detection of CPV by Real-Time PCR

In total, 46.6% (70/150) of all faecal samples showed positive qPCR results. In group H (healthy dogs), 24.0% (12/50), in group S (shelter dogs), 18.0% (9/50), and in group P (dogs suspected to have parvovirosis), 98.0% (49/50) of faecal samples were qPCR-positive. The mean CPV virus load was 2.4 × 10^6^ copies/g faeces and 2.7 × 10^6^ copies/g faeces for all qPCR-positive dogs in group H and S, respectively, and 5.3 × 10^13^ for all qPCR-positive dogs in group P.

### 3.2. Practicability of the Point-of-Care Tests

All eight POCT were easy to perform. In two POCT (Primagnost^®^ Parvo H + K and FASTest^®^ PARVO Card), faecal samples had to be collected using plastic spiral sticks that had to be introduced several times at various points in the faeces. The cotton swabs of the other tests had to be inserted only one time. There was no difference in the practicability of the tests. There were, however, some differences in the interpretation. The control fields of all POCT could always be clearly detected. Primagnost^®^ Parvo H + K (POCT-C) and FASTest^®^ PARVO Card (POCT-D) showed clear positive and negative results in all samples tested. However, the interpretation of some positive results in the other POCT was difficult due to very light test bands or points (Table 2): One WITNESS^®^ Parvo (POCT-H), two IDEXX Snap^®^ Parvo (POCT-A), two ImmunoRun^®^ Parvovirus Antigen Detection Kit (POCT-G) tests results, three Fassisi^®^ Parvo (POCT-B) test results, four Vetexpert Rapid Test CPV Ag^®^ (POCT-E) test results, and five Anigen Rapid CPV Ag Test Kit^®^ (POCT-F) test results were difficult to interpret.

### 3.3. Sensitivity, Specificity and Predictive Values of CPV by Point-of-Care Tests

The results of the sensitivity, specificity, predictive values and overall accuracy tested by the eight POCT compared to qPCR are shown in Table 2 and Table 3.

The specificity was excellent for all eight POCT (100.0% in all tests), as was the PPV (100.0% in all tests). There were no false positive results.

The Vetexpert Rapid Test CPV Ag^®^ (POCT-E) and Anigen Rapid CPV Ag Test Kit^®^ (POCT-F) had the highest sensitivity (34.3% and 32.9%, respectively) and highest NPV (63.5% and 63.0%, respectively). The Primagnost^®^ Parvo H + K (POCT-C) and FASTest^®^ PARVO Card (POCT-D) showed the lowest sensitivity (24.3% and 22.9%, respectively) and lowest NPV (60.2% and 59.7%, respectively).

The sensitivity and specificity of the POCT results within the three groups (healthy dogs, shelter dogs, CPV infection suspected dogs) are shown in Table 4. As none of the dogs of group H and S were tested positive in any of the POCT, all qPCR-positive faecal samples of group H, as well as all qPCR-positive faecal samples of group S were false negative in every POCT. In group P, the sensitivity of the POCT varied between 32.7% (FASTest^®^ PARVO Card (POCT-D)) and 49.0% (Vetexpert Rapid Test CPV Ag^®^ (POCT-E)).

### 3.4. Comparison of the Point-of-Care Tests

The differences among the POCT were found to be significant in group comparisons (*p* = 0.0003). The significant differences in sensitivities (shown in Table 5) were detected between the Vetexpert Rapid Test CPV Ag^®^ (POCT-E) and Fassisi^®^ Parvo (POCT-B) (McNemar’s *p*-value: 0.04), as well as between the Primagnost^®^ Parvo H + K (POCT-C) (McNemar’s *p*-value: 0.02) and FASTest^®^ PARVO Card (POCT-D) (McNemar’s *p*-value: 0.01). The sensitivity of the Anigen Rapid CPV Ag Test Kit^®^ (POCT-F) was significantly different from that of the Primagnost^®^ Parvo H + K (POCT-C) (McNemar’s *p*-value: 0.04) and FASTest^®^ PARVO Card (POCT-D) (McNemar’s *p*-value: 0.02). The IDEXX Snap^®^ Parvo (POCT-A) and FASTest^®^ PARVO Card (POCT-D) were also found to differ significantly in sensitivity (McNemar’s *p*-value: 0.04).

The agreement of the POCT results is shown in Table 5. For the Fassisi^®^ Parvo (POCT-B) and Anigen Rapid CPV Ag Test Kit^®^ (POCT-F); the Primagnost^®^ Parvo H + K (POCT-C) and Vetexpert Rapid Test CPV Ag^®^ (POCT-E), as well as the FASTest^®^ PARVO Card (POCT-D) and Anigen Rapid CPV Ag Test Kit^®^ (POCT-F) the agreement was substantial. For all other POCT, the agreement was almost perfect with Kappa values higher of than 0.80. The overall consistency of the POCT results in pairwise comparison of the eight evaluated test kits varied between 94.7% and 99.3%.

### 3.5. Detection Probability and Virus Culture

A regression analysis showed a significant dependence on the detection probability of the eight POCT on the virus load with *p* values < 0.001. A decision threshold (in the number of virus copies/g faeces) for every POCT was calculated. The threshold identified the necessary amount of virus copies/g faeces to reach a detection probability of 50% (50% correctly identified positive faecal samples). Decision thresholds varied between 1.406 × 10^12^ virus copies/g faeces for the POCT with the highest sensitivity (Vetexpert Rapid Test CPV Ag^®^ (POCT-E)) and 1.931 × 10^13^ for the POCT with the lowest sensitivity (FASTest^®^ PARVO Card (POCT-D)). The curves of all POCT with decision thresholds are shown in Figure 1.

A viral replication in culture was not observed in any faecal sample of group H (healthy dogs) and S (shelter dogs). In group P (dogs suspected to have parvovirosis), faecal culture was positive in 36.0% (18/50) of the faecal samples, and thus in 36.7% of the qPCR-positive samples. The results of virus culture can be seen in Appendix A. The virus culture-positive faecal samples had significantly higher viral loads compared to those in virus culture-negative faecal samples (*p* < 0.05) (Figure 2).

Sensitivities of the eight POCT of qPCR-positive and culture-positive faecal samples were between 83.3 and 94.4%; those of qPCR-positive and culture-negative faecal samples were between 1.9 and 9.5% (Table 6).

## 4. Discussion

CPV infection causes an acute and severe disease with a high mortality. Therefore, a fast and correct diagnosis is extremely important. The reference standard qPCR has to be performed in specialized laboratories and results are available generally no earlier than after a few days. Thus, POCT for in-house testing are an important measure to diagnose infected dogs immediately and directly at the veterinarian. Therefore, the aim of this study was to evaluate eight POCT regarding practicability, sensitivity, specificity, NPV, PPV, and overall accuracy when compared to qPCR.

The current study showed an excellent specificity of 100.0% of all eight POCT, with no false positive test results in all POCT. However, it has to be considered, that only dogs not vaccinated against CPV during the last four weeks, were included in groups H and S. A previous study showed the shedding of field and vaccination strains of CPV in 23.0% of dogs up to 28 days after modified-live vaccination, detected by qPCR [22]. This could lead to positive POCT results in the post-vaccination period, although in previous studies no positive POCT results after vaccination were observed despite high viral titres [14,17,23]. Three dogs of group P were vaccinated against CPV during the last four weeks. As described above, this could potentially have led to false positive test results in POCT and qPCR. All dogs of group P showed clinical signs of an infection with CPV. Consequently, it was assumed that these three dogs were not only shedding the vaccination virus but were infected with CPV and therefore tested correctly positive. 

All eight POCT used in this study were easy to perform and practicability was comparable for all of them. However, the overall sensitivity of the POCT for all dogs (group H, S and P) was very low (between 22.9% and 34.3%) with the Vetexpert Rapid Test CPV Ag^®^ (POCT-E) being the test with the highest sensitivity, and the FASTest^®^ PARVO Card (POCT-D) being the test with the lowest sensitivity. For the test performance, sensitivity (correctly positive-identified faecal samples) was considered as the most important parameter to avoid virus spread by unrecognized infected dogs and to provide intensive treatment to every dog with a symptomatic CPV infection. It must be considered that the POCT were only performed once and not repeated on the same fecal sample which, however, is the normal situation in the field where POCT are not repeated. In former studies, the sensitivity of the POCT showed great variability with values between 15.8% and 80.4% [12,14,16,17]. The reasons for the variability of sensitivities might be differences in the virus load of the faecal samples. Some former studies [12,16] did not rule out the correlation between POCT results and virus load which limits the comparability. In the present study, the detection probability of all POCT depended on the virus load. The lower the number of virus copies/g faeces, the lower the detection probability of the POCT. Therefore, a correlation between false negative POCT results and low virus loads was clearly shown in the present study which was in line with the results from Decaro and colleagues (2010) [17] and Proksch and colleagues (2015) [15]. Proksch and colleagues (2015) [15] showed that 51.3% of the IDEXX Snap^®^ Parvo test results compared to qPCR were false negative, with the false negative faecal samples in POCT having significantly lower viral loads. Decaro and colleagues (2010) [17] demonstrated that the IDEXX Snap^®^ Parvo had a good sensitivity of 80.4%, 78.0% and 77.0% for the detection of CPV 2-a, -b and -c compared to qPCR. However, the relatively high sensitivity rate in that study was likely the result of the preselection of samples to only include those containing high viral loads (>10^8^ DNA copies/g faeces). In the present study, the mean virus load in group H and S was 10^6^ DNA copies/g faeces and consequently significantly lower than the one in the preselected samples of Decaro and colleagues (2010) [17]. All qPCR-positive faecal samples of groups H and S were tested as false negative in every POCT. Therefore, the present study confirmed that samples with low virus loads could not be detected by POCT, only by qPCR. However, it must be mentioned that none of the samples of group H and S contained intact and infectious CPV in virus culture.

The overall low sensitivity of all POCT in the present study was partially caused by the high number of false negative results in groups H and S, presumably due to low virus loads. The dogs from both groups were healthy and showed no gastrointestinal signs. Schmitz and colleagues (2009) [16] also evaluated POCT with faeces from different groups of dogs, including dogs with acute hemorrhagic diarrhea, chronic hemorrhagic diarrhea and without gastrointestinal signs. The sensitivities of the POCT in this study were comparable to the results of the present study, with 18.4% for the IDEXX Snap^®^ Parvo, 15.8% for the FASTest^®^ PARVO Strip and 26.3% for the WITNESS^®^ Parvo card, and false negative POCT results also continuously occurred in the group of dogs without gastrointestinal signs, as well as in dogs with chronic haemorrhagaic diarrhea. The testing of clinically healthy animals was performed to identify asymptomatic virus shedders, for example in animal shelters [24]. The results of the present study showed that POCT were not useful at least in healthy dogs.

It was remarkable that 24.0% of the dogs in group H and 18.0% of the dogs in group S were qPCR-positive, respectively. None of these dogs were vaccinated against CPV within the last four weeks, and all dogs were healthy without any clinical signs. In a study by Freisl and colleagues (2017) [22], CPV shedding was also detected in healthy, client-owned dogs, but their prevalence was lower with 2.0%. Subclinical infections are the most likely reason for these positive PCR results. The role of subclinically infected dogs is not clear yet. Bergmann and colleagues (2019) [25] found FPV and CPV field virus DNA in clinically healthy cats. In that study, virus replication in virus culture showed that these cats shed an intact and infectious form of the virus and this is most likely possible for clinically asymptomatic dogs as well. With shedding infectious virus, these dogs might infect other susceptible animals and cause environmental contamination. However, in the present study, virus growth in virus cultures and, consequently, the excretion of the infectious field virus could not be detected for any of the asymptomatic dogs of group H and S. It must be considered that the qPCR detects viral DNA and not infectious viruses. Therefore, the role of positive qPCR results in asymptomatic dogs, as well as the importance of these dogs for environmental contamination should further be clarified in future studies.

In practice, dogs that are being tested by the veterinarian for CPV infection commonly show gastrointestinal signs. Thus, in the present study, group P represents the group, where POCT are most commonly used. In this group, 98.0% of the faecal samples were qPCR-positive and, consequently, these dogs suffered from a clinically relevant parvovirosis. The mean virus load of this group was 10^13^ DNA copies/g faeces and notably higher than that in groups H and S (10^6^ DNA copies/g faeces). The results of the virus culture showed a growth of intact CPV in 36.0% of all faecal samples of group P. The sensitivities of the POCT using only samples from group P were between 32.7 and 49.0%. This is still much too low and comparable with the results of a study from Proksch and colleagues (2015) evaluating a POCT only with samples from dogs with confirmed parvovirosis [16].

A reason for the false negative POCT results in group P could be the time of testing. Experimental studies showed that CPV shedding started 3–4 days after inoculation [26,27] with the highest virus shedding 4–7 days after inoculation [5,26]. As qPCR was able to detect lower virus loads, infections in the early and late stages might only be recognized by qPCR, but not by POCT. Despite the high viral loads, some faecal samples of group P tested negative in all POCT. One possible explanation for this finding might be the consinstency of the faeces of these samples. Their texture was very bloody with significant mucous membrane particles which could cause detection problems. In addition, POCT detected virions, whereas qPCR detected DNA particles; therefore, infected cells might contain more DNA particles than virions [14]. Another reason for false negative POCT results could be high titres of interfering antibodies, which sequestrate viral antigen. Thus, the antigen might not be detected by POCT [15,17]. This could explain, why some faecal samples of group P in this study were not detected by any POCT in spite of high viral loads and positive faecal virus cultures.

One limitation is that all POC tests were only performed once. Thus, operating errors, which could have been minimized with multiple test runs, must be taken into consideration. Accordingly, the sensitivity and specificity of the POCT, as reported in the present study could be subject to inter-operator variability and could deviate in clinical practice depending on the exact operating conditions. Therefore, the sensitivities of the POCT determined in the present study represented only a general statement and no absolute rating. However, POCT were also generally not repeated in veterinary practice, and thus the present study mimics the practice situation in which the POCT are used. Larger clinical studies including the analysis of inter-operator and inter-reader variability would be useful to ascertain the clinical accuracy of the POCT.

## 5. Conclusions

In conclusion, all POCT were easy to perform and were suitable as screening tests in veterinary practices and clinics. A positive POCT result confirmed an infection with CPV in dogs not vaccinated against CPV within the last weeks. However, due to the low sensitivities and following false negative test results in all POCT, a negative POCT result could not definitely rule out a parvovirus infection. In this study, the sensitivity of all POCT in group P ranged from 32.7% to 49.0%. Based on these results, in the case of a negative POCT, a confirmatory PCR test should be performed in dogs showing clinical signs of CPV infection. In healthy dogs, POCT are not useful. Although the Vetexpert Rapid Test CPV Ag^®^ (POCT-E) showed the highest sensitivity compared to the other POCT, the sensitivity of all POCT was too low and must be improved to avoid false negative test results. The qPCR was able to detect very small amounts of viral DNA and, consequently, detect subclinical infections. The importance of a positive qPCR result in clinically healthy dogs still has to be clarified.

## Figures and Tables

**Figure 1 viruses-13-02080-f001:**
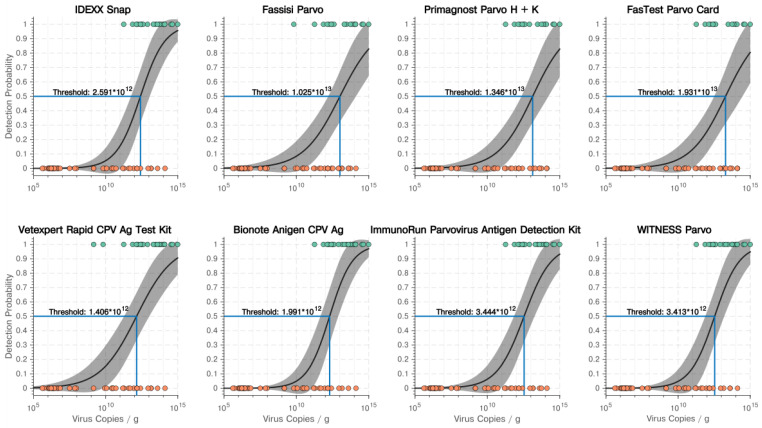
Results of the regression analysis showing the dependence of the detection probability on the virus load with decision thresholds (detection probability of 50%) for all eight point-of-care tests. Green points indicate correct positive point-of-care test results, red points false negative point-of-care test results.

**Figure 2 viruses-13-02080-f002:**
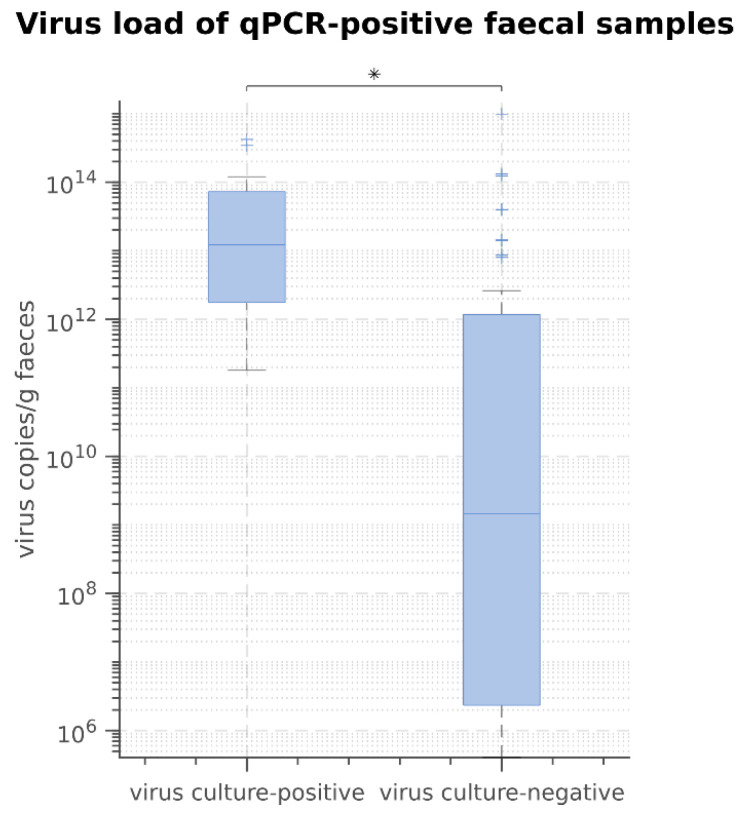
Results of comparison of viral loads in qPCR-positive samples between virus culture-positive and virus culture-negative faecal samples using Mann–Whitney-U-Test. * *p* = 0.01.

**Table 1 viruses-13-02080-t001:** Eight point-of-care tests for the detection of canine parvovirus in faeces and the respective manufacturers’ instructions.

POCT	Product	Manufacturer	Storage Requirements	Usage	Duration of Test Performance	Test Material	Reagent Tray	Price per Test in Germany (Excl. VAT)
POCT-A	Snap^®^ Parvo	IDEXX (Westbrook, USA)	+2–25 °C	After warming up to room temperature	8 min	Faeces	Cotton swab	EUR 13.08
POCT-B	Fassisi^®^ Parvo	Fassisi (Göttingen, Germany)	+2–30 °C	After warming up to room temperature	10 min	Faeces	Cotton swab	EUR 4.20
POCT-C	Primagnost^®^ Parvo H + K	Dechra (Aulendorf, Germany)	+15–25 °C	At room temperature	5 min	Faeces	Plastic spiral stick	EUR 6.51
POCT-D	FASTest^®^ PARVO Card	Megacor (Lindau, Germany)	+15–25 °C	At room temperature	5 min	Faeces	Plastic spiral stick	EUR 5.42
POCT-E	Vetexpert Rapid Test CPV Ag^®^	Vetexpert (Vienna, Austria)	+2–30 °C	After warming up to room temperature	5–10 min	Faeces	Cotton swab	EUR 5.36
POCT-F	Anigen Rapid CPV Ag Test Kit^®^	Bionote (Dongtan, South Korea)	+2–30 °C	After warming up to room temperature	10 min	Faeces	Cotton swab	EUR 4.45
POCT-G	ImmunoRun^®^ Parvovirus Antigen Detection Kit	Biogal (Galed, Israel)	+2–30 °C	After warming up to room temperature	5–10 min	Faeces	Cotton swab	EUR 6.39
POCT-H	WITNESS^®^ Parvo	Zoetis (Parsippany, USA)	+2–25 °C	After warming up to room temperature	5 min	Faeces	Cotton swab	EUR 10.83

POCT, point-of-care tests, VAT, value added tax, EUR, euro.

**Table 2 viruses-13-02080-t002:** Performance parameters of the eight point-of-care tests to detect canine parvovirus antigen in faeces when considering all 150 samples: tests which were difficult to interpret, sensitivity, specificity, positive and negative predictive values, as well as overall accuracy, were calculated using real-time polymerase chain reaction as gold standard.

Tests	POCT-A	POCT-B	POCT-C	POCT-D	POCT-E	POCT-F	POCT-G	POCT-H
Tests difficult to interpret %	1.3	2.0	0.0	0.0	2.7	3.3	1.3	0.7
(number of tests/total)	(2/150)	(3/150)	(0/150)	(0/150)	(4/150)	(5/150)	(2/150)	(1/150)
Sensitivity %	31.4	25.7	24.3	22.9	34.3	32.9	30.0	30.0
(95% CI)	(24.2–39.5)	(19.1–33.5)	(17.8–32.0)	(16.6–30.5)	(26.9–42.5)	(25.5–41.0)	(22.9–38.1)	(22.9–38.1)
Specificity %	100.0	100.0	100.0	100.0	100.0	100.0	100.0	100.0
(95% CI)	(96.9–100)	(96.9–100)	(96.9–100)	(96.9–100)	(96.9–100)	(96.9–100)	(96.9–100)	(96.9–100)
Positive predictive value %	100.0	100.0	100.0	100.0	100.0	100.0	100.0	100.0
(95% CI)	(96.9–100)	(96.9–100)	(96.9–100)	(96.9–100)	(96.9–100)	(96.9–100)	(96.9–100)	(96.9–100)
Negative predictive value %	62.5	60.6	60.2	59.7	63.5	63.0	62.0	62.0
(95% CI)	(54.2–70.2)	(52.3–68.4)	(51.8–68.0)	(51.4–67.5)	(55.2–71.1)	(54.7–70.6)	(53.7–69.7)	(53.7–69.7)
Overall accuracy %	68.0	65.3	64.7	64.0	69.3	68.7	67.3	67.3
(95% CI)	(59.8–75.3)	(57.1–72.8)	(56.4–72.2)	(55.7–71.6)	(61.2–76.5)	(60.5–75.9)	(59.1–74.7)	(59.1–74.7)

qPCR, real-time polymerase chain reaction, POCT, point-of-care test, CI, confidence interval.

**Table 3 viruses-13-02080-t003:** Results of eight canine parvovirus point-of-care tests of 150 faecal samples in comparison with real-time polymerase chain reaction as gold standard.

	POCT-A Negative	POCT-A Positive		POCT-B Negative	POCT-B Positive
**qPCR-negative** ** *n* ** **= 80**	80 true negative	0 false positive	**qPCR-negative** ** *n* ** **= 80**	80 true negative	0 false positive
**qPCR-positive** ** *n* ** **= 70**	48 false negative	22 true postive	**qPCR-positive** ** *n* ** **= 70**	52 false negative	18 true postive
**Total**	128	22	**Total**	132	18
	**POCT-C** **Negative**	**POCT-C** **Postive**		**POCT-D** **Negative**	**POCT-D** **Positive**
**qPCR-negative** ** *n* ** **= 80**	80 true negative	0 false positive	**qPCR-negative** ** *n* ** **= 80**	80 true negative	0 false positive
**qPCR-positive** ** *n* ** **= 70**	53 false negative	17 true postive	**qPCR-positive** ** *n* ** **= 70**	54 false negative	16 true postive
**Total**	133	17	**Total**	134	16
	**POCT-E** **Negative**	**POCT-E** **Positive**		**POCT-F** **Negative**	**POCT-F** **Postive**
**qPCR-negative** ** *n* ** **= 80**	80 true negative	0 false positive	**qPCR-negative** ** *n* ** **= 80**	80 true negative	0 false positive
**qPCR-positive** ** *n* ** **= 70**	46 false negative	24 true postive	**qPCR-positive** ** *n* ** **= 70**	47 false negative	23 true postive
**Total**	126	24	**Total**	127	23
	**POCT-G** **Negative**	**POCT-G** **Positive**		**POCT-H** **Negative**	**POCT-H** **Positive**
**qPCR-negative** ** *n* ** **= 80**	80 true negative	0 false positive	**qPCR-negative** ** *n* ** **= 80**	80 true negative	0 false positive
**qPCR-positive** ** *n* ** **= 70**	49 false negative	21 true postive	**qPCR-positive** ** *n* ** **= 70**	49 false negative	21 true postive
**Total**	129	21	**Total**	129	21

qPCR, real-time polymerase chain reaction, POCT, point-of-care test.

**Table 4 viruses-13-02080-t004:** Sensitivity and specificity of eight canine parvovirus point-of-care tests to detect canine parvovirus antigen in faeces in comparison with real-time polymerase chain reaction as gold standard for three groups (each 50 dogs).

	Tests	POCT-A	POCT-B	POCT-C	POCT-D	POCT-E	POCT-F	POCT-G	POCT-H
Group H real-time PCR-positive: 24.0% (12/50) mean virus load: 2.4 × 10^6^	Sensitivity %	0.0	0.0	0.0	0.0	0.0	0.0	0.0	0.0
	Specificity %	100.0	100.0	100.0	100.0	100.0	100.0	100.0	100.0
Group S real-time PCR-positive: 18.0% (9/50) mean virus load: 2.7 × 10^6^	Sensitivity %	0.0	0.0	0.0	0.0	0.0	0.0	0.0	0.0
	Specificity %	100.0	100.0	100.0	100.0	100.0	100.0	100.0	100.0
Group P real-time PCR-positive: 98.0% (49/50) mean virus load: 5.3 × 10^13^	Sensitivity %	44.9	36.7	34.7	32.7	49.0	46.9	42.9	42.9
	Specificity %	100.0	100.0	100.0	100.0	100.0	100.0	100.0	100.0

POCT, point-of-care test, qPCR, real-time polymerase chain reaction; Group H: healthy, client-owned dogs, not vaccinated against canine parvovirus during the last four weeks. Group S: healthy shelter dogs, not vaccinated against canine parvovirus during the last four weeks. Group P: dogs suspected to be infected with canine parvovirus.

**Table 5 viruses-13-02080-t005:** Results of McNemar’s statistic to determine differences in sensitivity of the eight point-of-care tests detecting canine parvovirus antigen in faeces with overall consistency and kappa coefficent of the point-of-care test results in direct comparison.

			Overall Consistency in % Kappa Coefficent (κ)							
	Sensitivity in %		POC-A	POC-B	POC-C	POC-D	POC-E	POC-F	POC-G	POC-H
Mc Nemar’s *p*-value for pairwise comparison of sensitivities	31.4	POC-A		96.00 κ = 0.83	96.67 κ = 0.85	96.00 κ = 0.82	98.67 κ = 0.95	99.33 κ = 0.97	99.33 κ = 0.97	98.00 κ = 0.92
	25.7	POC-B	0.22		96.67 κ = 0.84	97.33 κ = 0.87	96.00 κ = 0.83	95.33 κ = 0.80	96.67 κ = 0.85	96.67 κ = 0.85
	24.3	POC-C	0.07	1.00		99.33 κ = 0.97	95.33 κ = 0.80	96.00 κ = 0.83	97.33 κ = 0.88	97.33 κ = 0.88
	22.9	POC-D	**0.04 ***	0.62	1.00		94.67 κ = 0.83	95.33 κ = 0.80	96.67 κ = 0.85	96.67 κ = 0.85
	34.3	POC-E	0.48	**0.04 ***	**0.02 ***	**0.01 ***		98.00 κ = 0.92	98.00 κ = 0.92	96.67 κ = 0.87
	32.9	POC-F	1.00	0.13	**0.04 ***	**0.02 ***	1.00		98.67 κ = 0.95	98.67 κ = 0.95
	30.0	POC-G	1.00	0.37	0.13	0.07	0.25	0.48		97.33 κ = 0.89
	30.0	POC-H	1.00	0.37	0.13	0.07	0.37	0.48	0.62	

POCT, point-of-care test; * bold values indicate significant difference (*p* ≤ 0.05); κ values < 0 indicate poor agreement, 0.00–0.20 slight, 0.21–0.40 fair, 0.41–0.60 moderate, 0.61–0.80 substantial, and 0.81–1.00 almost perfect agreement.

**Table 6 viruses-13-02080-t006:** Sensitivities of the eight point-of-care-tests in groups of real-time polymerase chain reaction-positive, virus culture-positive, real-time polymerase chain reaction-positive, and virus culture-negative faecal samples.

	POCT-A	POCT-B	POCT-C	POCT-D	POCT-E	POCT-F	POCT-G	POCT-H
Sensitivity culture-positive faecal samples (in %)	94.4	83.3	83.3	83.3	83.3	94.4	88.9	94.4
Sensitivity culture-negative faecal sample (in %)	9.6	5.8	3.8	1.9	13.5	11.5	9.6	7.7

POCT, point-of-care test.

## Data Availability

The authors confirm that the datasets analyzed during the study are available from the first and corresponding author upon reasonable request.

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
