# Peer review of "Comparison of Eight Commercially Available Faecal Point-of-Care Tests for Detection of Canine Parvovirus Antigen"

_viruses, 2021, doi:10.3390/v13102080_

Round 1
Reviewer 1 Report
Journal: Viruses
Manuscript ID: viruses-1383983
Title: Comparison of eight commercially available faecal point-of-care tests for detection of canine parvovirus antigen
- Overview and general recommendation:
In this study, Walter-Weingärtner and co-Authors evaluated eight faecal point-of-care tests (POCT) for detection of canine parvovirus antigen, analyzing 150 fecal samples collected from healthy and symptomatic dogs. Results were compared to those obtained from the real-time PCR assays. Six different parameters were considered in the evaluation. Authors observed high specificity but low sensibility for all the tested POCT and commented these results comparing with those of previous studies.
This study represents a valid insight in the current literature and these data could have a clinical and perspective relevance. Moreover, this study is well written, with a complete description.
I added some comments and suggestions to Authors to improve the description throughout the manuscript. I explained these comments and suggestions in more details below.
2.1 Major comments:
- Lines 22-25: Authors included in the abstract only the sensitivity values referring to the entire sample under examination. Only in the following paragraphs the sensitivity values referring to the group P is described and commented. Due to the clinical relevance of this group, I suggest trying to include these results in the abstract.
- Line 27: A negative POCT result, if true negative, excludes CPV infection. Thus, generically the negative POCT result potentially does not exclude definitely CPV infection. I suggest reconsidering this potential misunderstanding.
- Par.2.1: It is not clear the age of the dogs within the three groups and information on the previous vaccinations are missing. Indeed, dogs in groups H and S were not vaccinated against CPV within the last four weeks and, thus, the interference of vaccine shedding was likely excluded but the same information is not clear for dogs in group P. Again, a previous vaccination in groups H and S could have triggered an immune response that would contribute to the results obtained. Following these considerations, I suggest to clarify, whatever available, these details.
Moreover, it is not clear the mean of “incorrect CPV vaccination status”: were these dogs incorrectly, improperly vaccinated or their vaccination was incomplete according to a specific standard (i.e., WSAVA or other international veterinary association guidelines)? I suggest to clarify these details.
- Line 148: was it difficult the interpretation of some positive results related to the lower virus loads?
- Table 7: following the previous comment, were some samples (i.e., numbers 148, 121, 111) difficult to interpreter? Indeed, i.e., sample #148 showed a similar virus load to sample #135 but clearly different POCT results or samples #121 and 111 showed higher viral loads compared to sample #135 but virus isolation in cell cultures and, particularly, POCT results were different. How Authors explained these results?
- Line 252: following the first comment, I suggest to specify that the discussed results in this part are referred to the entire sample under examination.
- Line 259-261: In some cited studies (i.e., references 13 and 15), the correlation between POCT results and virus load was not ruled out and this limit the comparison. In this study, a clearer correlation between false negative POCT results and low virus loads was more clearly evidenced and is in line with data in Decaro et al., 2010 (reference 17) and in Proksch et al., 2015 (reference 14).
- Line 271: This sentence is not clear: despite this situation does not necessarily represents what happens in the field, higher viral loads are common in faeces of dogs infected with CPV and showing clinical signs. To avoid any misinterpretation of this sentence, I suggest to clarify it. Moreover, in reference 17, “samples of the different CPV types containing viral loads >105 DNA copies/mg faeces” were selected for the study.
- Line 332-334: as above observed, a potential false negative POCT result in dogs showing clinical signs of suspect parvovirosis usually lead to the repetition of the POCT or, more correctly, to test samples by PCR-based tests. In this study, sensitivity of all PCT in group P ranged from 32.7 to 49.0 (as in Table 4 and lines 177-178). Based on these results, I suggest to emphasize the need for further tests in this specific cases (dogs suspected to have parvovirosis) rather than in healthy dogs to avoid misunderstanding on the advantages or limits of POCT in the small animal clinical practice. A similar observation was made by Schmitz et al., at reference 15.
2.2 Minor comments:
- Line 13: diagnosis is a clinical judgment which consists in recognition of a morbid condition based on the clinical examination and on laboratory and instrumental exams. Following this interpretation, I suggest to add “laboratory” before “diagnosis” or change “for” with “to confirm the”. Something similar could be also used to complete this first sentence.
- Line 20: the comma after “(PPV)” could be removed. Similarly in the Introduction at line 61.
- Line 37: I suggest to rephrase with “Clinical signs and laboratory findings”.
- Lines 43-45: A reference could be added.
- Line 79: I suggest to replace “isolated” with “extracted”.
- Line 82: I suggest to move the reference number close to the citation (i.e., after “2013” at line 81). Similarly, the reference numbers at lines 267 (to line 265), 269 (to line 267), 284 (to line 282), 295 (to line 294), 299 (to line 297) could be moved close to their citations.
- Line 97: similarly, I suggest to move “(Table 1)” after “antigen” at line 96; Authors could also consider to add “details in” before “Table 1”. Moreover, were POCT results or sample testing with POCT not repeated?
- Table 1: Plus or minus Celsius degrees as storage temperature?
- Line 121: I suggest to use the plural noun for “performance”.
- Lines 178-179: as these results are related to those included in paragraph 3.1, it could be moved in the specific paragraph.
- Line 201: probably the caption of the Table 5 was merged to the text of the paragraph. Please, revise it.
- Lines 211-213: similarly to the previous comment, the caption of Table 6 was merged to the title of the paragraph 3.5. Please, revise it.
- Line 223: Please check if “was not observed” could replace “could not be detected”.
- Lines 223-228: I suggest to move this part between Figure 1 and Table 7.
- Line 274: I suggest to add “[17]” after “(2010)”.
- Lines 296 and 299: for both these reasons, Authors could suggest the need of further studies to elucidate these findings.
Author Response
Thank you for the review, reply can be found in the attachment.
Please see the attachment.

Reviewer 2 Report
Walter-Weingärtner et al. compared 8 commercially available rapid tests to diagnose CPV-2 infection to qPCR results (both qualitatively and quantitatively). All tests had 100% specificity, but sensitivity was low for all tests both in the overall population and in dogs with clinical signs of CPV-2 infection. There was a correlation between viral load, symptomatology, and sensitivity. The study is well performed and the manuscript well-written. These results are important because CPV-2 is still a big issue in dogs and an effort should be made to improve POC diagnostics for this pathogen. There are only a few points I think that maybe can be evaluated to make the study more complete.
- One aspect that is important to consider when dealing with these kinds of tests is costs, especially at a shelter. As all tests were easy to perform and they were all equally bad, maybe the cost could be a decisive factor. Maybe you can add this to table 1.
- Was there a difference in sensitivity of the 8 POCT in the 2 groups qPCR-positive/culture-positive and qPCR-positive/culture-negative? Also, was there a difference in viral load between these two groups (looking at table 7 I think there is)?
- What were the detection limits (in terms of viral load) or each POCT? Maybe these could be added to table 2?
- Tables. There are a lot of tables in this paper. I am wondering if you could combine Tables 5 and 6 (e.g. one parameter top-right and one bottom-left). Also, Table 7 is long, weirdly formatted, and not particularly informative and should be removed (the data is already in supplementary material anyway). Finally, a way to make all tables more streamlined is to remove the specific names of each test and just refer to them as A-H?
Minor:
Title and everywhere else in the text. The virus is normally referred to as Canine parvovirus type 2 (CPV-2). Please refrain from using CPV alone as it generates confusion.
Lines 31-2. It is currently believed that CPV-2 emerged from an FPV-like virus, not from FPV directly.
Section 2.2. How were tests with discordant results (those that were positive in one test and negative in the other test, if present) considered?
Line 94. How many days after the second passage was quantification performed?
Lines 165-7. How can you calculate these values if the qPCR is the gold standard? What did you compare your qPCR results to? I suggest removing this…
Lines 184-7, 201, 211-213. I think something went wrong with formatting here.
Lines 252-4. The repetition of the sensitivity values can be avoided.
Line 291. Remove the comma after remarkable.
Lines 316-320. However, the samples in these group all had high viral loads.
Line 326. I’d replace “further” with “until”.
Lines 327-8. Parvoviruses are extremely stable, and I find this explanation unlikely.
An explicit reference to supplementary material is missing from the text.
Maybe you can mention that practicability was comparable for all tests in the discussion to highlight that this aspect also doesn’t point in favor of any particular test.
Author Response
Thank you for the revision. Reply can be found in the attachment.
Please see the attachment.
